# Methodology for Testing Key Parameters of Array-Level Small-Area Hafnium-Based Ferroelectric Capacitors Using Time-to-Digital Converter and Capacitance Calibration Circuits

**DOI:** 10.3390/mi14101851

**Published:** 2023-09-27

**Authors:** Donglin Zhang, Honghu Yang, Yue Cao, Zhongze Han, Yixuan Liu, Qiqiao Wu, Yongkang Han, Haijun Jiang, Jianguo Yang

**Affiliations:** 1Institute of Microelectronics of Chinese Academy of Sciences, Beijing 100029, China; zhangdonglin20@mails.ucas.ac.cn (D.Z.); hanzhongze@ime.ac.cn (Z.H.); 2School of Microelectronics, University of Chinese Academy of Sciences, Beijing 100049, China; 3School of Microelectronics, University of Science and Technology of China, Hefei 230026, China; yanghonghu@mail.ustc.edu.cn; 4School of Microelectronics, Fudan University, Shanghai 200433, China; caoyue@zhejianglab.com (Y.C.); 20112020109@fudan.edu.cn (Y.L.); wuqiqiao@mail.ustc.edu.cn (Q.W.); 5Zhangjiang Lab, Shanghai 201210, China; hanyk@zhejianglab.com (Y.H.); jianghaijun@zhejianglab.com (H.J.)

**Keywords:** ferroelectric capacitor, hafnium, test methodology, small area, array, non-ideal effect, cap calibration, VTC, TDC

## Abstract

Hafnium-based ferroelectric memories are a promising approach to enhancing integrated circuit performance, offering advantages such as miniaturization, compatibility with CMOS technology, fast read and write speeds, non-volatility, and low power consumption. However, FeRAM (Ferroelectric Random Access Memory) still faces challenges related to endurance and retention susceptibility to process variations. Hence, testing and obtaining the core parameters of ferroelectric capacitors continuously is essential to investigate these phenomena and explore the potential solution. The traditional method for measuring ferroelectric capacitors has limitations in timing generation capability, introduces parasitic capacitance, and lacks accuracy for small-area capacitors. In this study, we analyzed the working principle of ferroelectric capacitors and designed a method to detect the remnant polarization, saturation polarization, and imprint offset of ferroelectric capacitors. Further, we further proposed a circuit implementation method. The proposed test circuit conquers these limitations and enables high-precision testing of ferroelectric capacitors, contributing to developing hafnium-based ferroelectric memories. The circuit includes a flip-readout circuit, a capacitance calibration circuit, and a voltage-to-time converter and time-to-digital converter (VTC&TDC) readout circuit. According to simulation results, the capacitance calibration circuit reduces the deviation of the capacitance by 84%, and the accuracy of the readout circuit is 5.91 bits, with a readout time of 150 ns and a power consumption of 1 mW. This circuit enables low-cost acquisition of array-level small-area ferroelectric capacitance data, which can guide subsequent device optimization and circuit design.

## 1. Introduction

With the gradual end of Moore’s Law, it has become challenging to further improve the performance of integrated circuits by reducing line widths. This challenge has prompted researchers to seek new methods to enhance integrated circuit performance. One such approach is the development of novel devices, with new types of memories being a critical direction. Hafnium-based ferroelectric memory stands out as a promising candidate among these emerging memory technologies [1,2,3,4,5,6,7,8,9,10].

Ferroelectric materials were extensively studied, and ferroelectric memories based on traditional materials, such as PZT or SBT, were commercialized around 2000, offering products ranging from kilobits to megabits. However, due to material limitations, their miniaturization was challenging, making them cost-ineffective compared to mainstream memory technologies. Consequently, traditional ferroelectric memories gradually became marginalized.

In 2011, hafnium-based ferroelectric materials emerged [11], bringing about a renaissance in ferroelectric memory. These materials enable effective miniaturization, with film thicknesses below 5 nm. Hafnium-based ferroelectric memories are also compatible with CMOS technology, only requiring adding two extra masking layers in the conventional CMOS process, significantly reducing costs. In addition to the substantial cost advantages over traditional materials, hafnium-based ferroelectric memories offer several benefits. First, they exhibit fast read and write speeds, comparable to DRAM, achieving approximately 20 ns. Second, they possess non-volatility, allowing data retention without regular refreshing, while consuming significantly lower power than DRAM.

However, hafnium-based ferroelectric memories still face challenges. Their endurance is relatively low, with read and write cycles in the order of 10^9^ [12], while traditional ferroelectric memories can achieve cycles in the order of 10^15^ [13] and DRAM can reach cycles in the order of 10^16^. Moreover, hafnium-based ferroelectric memories exhibit high coercive fields, making them susceptible to fluctuations due to temperature and process variations, which impacts their practical application [14,15,16,17].

To investigate these issues, researchers made extensive tests of the devices to obtain critical parameters such as remnant polarization, saturation polarization, and coercive field offset. These parameters are crucial for evaluating device performance. The traditional method for measuring ferroelectric capacitors involves using semiconductor analyzers or specialized ferroelectric analyzers, as depicted in Figure 1. The process requires probing the metal pads connected to the device and generating pulse signals subsequently to obtain the P–V curve of the ferroelectric capacitor. Although these instruments enable easy access to the parameters of ferroelectric capacitors, they have significant limitations.

Firstly, the equipment is highly expensive with substantial testing costs. Secondly, the equipment has limited timing capabilities, only generating low-speed pulses, which do not meet the testing requirements for high-speed read and write operations of ferroelectric capacitors. Thirdly, the equipment requires the use of probes to connect to the pad or IO interface of the test chip, introducing a significant amount of parasitic capacitance. Moreover, for small-area ferroelectric capacitors, such as 0.7 μm × 0.7 μm, the charge generated is at the femtocoulomb level, which is difficult to achieve with such high accuracy using the equipment system. 

Additionally, the results obtained from testing large-area ferroelectric capacitors (100 μm^2^) cannot be directly applied to small-area ferroelectric capacitors (0.5 μm^2^) due to condition differences. The critical parameters of ferroelectric capacitance are closely related to the height and width of the applied pulses. When applying small-area ferroelectric capacitors in low-voltage and high-speed scenarios, the data obtained from testing large-area ferroelectric capacitors under high-voltage and low-speed conditions cannot be directly applied. Further, no literature directly indicates that the properties of large-area ferroelectric capacitors are identical to those of small-area ferroelectric capacitors, creating a research gap. Obtaining the relevant properties of small-area ferroelectric capacitors is an essential prerequisite for advancing the miniaturization of ferroelectric capacitors.

Furthermore, many properties of ferroelectric capacitors exhibit a significant distribution, such as remanent polarization and read/write times. Traditional testing methods are cumbersome, slow, and often limited to testing only a few dozen devices. This cannot provide a comprehensive distribution and may lead to the inability to measure certain tail bits or significant testing result deviations. Array-level, high-volume testing is required.

Therefore, we ought to design a specialized high-precision test circuit for on-chip testing of small-area hafnium-based ferroelectric capacitors to obtain crucial parameters. 

This work comprises three main parts. The first part introduces the background and motivation related to the research. In the second part, we comprehensively explain the charge transfer principle of ferroelectric capacitors and propose a methodology for testing key parameters. The third part presents the working principle, specific design, and simulation results of the specialized ferroelectric capacitor test circuit.

## 2. Materials, Charge Transfer Principle and Testing Methodology

Ferroelectric capacitors are non-volatile capacitive devices that can store charge. Figure 2a shows the TEM image. From the image, it can be observed that the interface between the bottom electrode TiN and the substrate (Box) is clear and smooth. The interface between the top electrode TiN and the contact metal Pd is clear. The interface between HZO and the upper and lower TiN electrodes is clear, and smooth, and the thickness of HZO is approximately10 nm.

Subsequently, the ferroelectric tester was used to test the P–V curve of the prepared hafnium-based ferroelectric capacitor. The applied pulse waveform was a PUND waveform with a pulse height of ±3 V, pulse width of 100 μs, and pulse interval of 100 μs. The area of the tested ferroelectric capacitor was 50 μm × 50 μm. The waveform and the obtained ferroelectric capacitor P–V curve are shown in Figure 2a,b.

The test results showed that under a voltage scan of 3 V, the double remnant polarization value was 48 μC/cm^2^, and the polarization difference between the remnant polarization P1 and the saturation polarization P0sat was 58 μC/cm^2^. In subsequent simulations, we rely on the above test results, use MATLAB interpolation, and fit the curve for further testing and simulations. In the following sections, to better illustrate the characteristics of small-area ferroelectric capacitors, we choose the charge quantity as the y-axis.

The simulation in this article involves two ferroelectric capacitor models: a mathematical model and a circuit model.

The mathematical model is used for data simulation to obtain design parameters. The measured data are initially divided into two monotonic datasets, which are then interpolated and extended. Subsequently, the fitting function tool in Matlab is employed to fit the monotonic datasets, resulting in two functions for subsequent data simulation calculations.

The ferroelectric capacitor switch model utilized in this study is based on the LK equation that Aziz A et al. proposed [18], optimized using experimental data shown in Figure 3a. A 3 V pulse-width-modulated signal with a frequency of 1 MHz was applied to the model. The test results are shown in Figure 3b.

The modeling approach in this work is consistent with [19]. C1 and R1 are parasitic parameters of the ferroelectric capacitor, where Vm represents the non-linear capacitance. Im converts voltage values into currents to charge the internal capacitor C2, which has a value of 1 F and is used to accumulate current. R2 is introduced as a bypass resistor to prevent convergence issues during calculations. The specific model parameter values are as shown in Table 1.

Vm is a voltage-controlled voltage source, and its expression is given by Vm = A × Vs + B × Vs^3^ + C × Vs^5^. Im is a current-controlled current source, expressed as Im = Vm.

Figure 4 represents the writing process of a ferroelectric capacitor for storing data “1”. Figure 4b illustrates the ferroelectric capacitor structure in an unpolarized state, where the internal domains are disordered. In this disordered state, the electric fields of the domains cancel each other out, resulting in no net polarization externally. The positive and negative charges on the electrodes are equal, and the electrodes are grounded. Figure 4c shows the ferroelectric capacitor in a negatively saturated polarization state. When a saturation voltage, *V_sat_*, is applied to the ferroelectric capacitor, charges immediately accumulate on both sides. These charges are called accumulated charges. Subsequently, an electric field is generated within the capacitor, causing the ferroelectric domains to flip under the influence of the electric field. After the flip, additional electric fields are generated, further attracting charges. These charges are called bound charges. The amount of bound charge is significantly larger than the accumulated charge. For example, a 100 μm^2^ area capacitor with a 130 nm process under a 3 V voltage could accommodate approximately 300 fC. However, the ferroelectric capacitor can attract approximately 30,000 fC of bound charge under the same conditions, which is 100 times larger than the accumulated charge. Therefore, for subsequent analysis, we neglect the accumulated charge. Figure 4d illustrates the ferroelectric capacitor in a negative remnant polarization state. At this point, the voltage across the ferroelectric capacitor is 0, and some domains spontaneously revert to the disordered state. In contrast, the rest retain their original state, resulting in the continued binding of charges by these domains. The amount of charge bound is denoted as *Q*_1_.

Figure 5 depicts the process of reading data from a ferroelectric capacitor. Figure 5b represents the state of storing data “1” in the ferroelectric capacitor. Initially, the voltage (*V_sat_*) is applied to the negative electrode of the ferroelectric capacitor, and charges flow onto the bit-line capacitor (*C_bl_*), resulting in the voltage (*V_BL_*_1_) on the bit-line capacitor. For small-area ferroelectric capacitors, their intrinsic capacitance is very small, so it is assumed that the voltage applied to the negative electrode falls entirely across the ferroelectric capacitor. At the moment of domain flipping, additional charges are extracted onto the bit-line capacitor, causing a change in potential. The applied voltage on the ferroelectric capacitor decreases, leading to a reduction in domain flipping. Eventually, the ferroelectric capacitor reaches a stable state, and the bound charge also stabilizes. At this point, the accumulated changes in bound charge are stored on the bit-line capacitor, resulting in the reading voltage *V_BL_*_1_ for data “1”. 

Due to charge conservation, the node connecting the positive electrode of the ferroelectric capacitor to the bit-line capacitor remains connected, and any changes in charge on the ferroelectric capacitor are transferred to the bit-line capacitor. The relationship can be expressed as follows:


(1)
$$\begin{array}{c} Q_{1}=-V_{BL}C_{BL}+Q\left(V_{fe}\right)\\ V_{BL}=V_{sat}-V_{fe}\\ V_{fe}=\frac{Q\left(V_{fe}\right)-Q_{1}}{C_{BL}}+V_{sat} \end{array}$$


When the capacitance of the bit-line capacitor is sufficiently large, as shown in Figure 5a, the slope becomes steep, and the voltage difference across the ferroelectric capacitor approaches the applied voltage, *V_sat_*, with *V_BL_* being very small. This condition represents the maximum polarization under the applied voltage. Thus, by selecting a sufficiently large capacitance, the voltage applied can be guaranteed to fall entirely across the ferroelectric capacitor, ensuring the polarization level of the ferroelectric capacitor. Consequently, the sampling of the bit-line capacitor can accurately represent the difference in charge between the negative remanent polarization and positive saturation polarization.

The hysteresis loop of a ferroelectric capacitor exhibits good symmetry. Based on this characteristic, a scheme for measuring the remnant polarization is designed. As shown in Figure 6, the ferroelectric capacitor is first written into the *Q*_1_ and *Q*_0_ states, followed by reading out to the saturation polarization points (*Q*_0*sat*_). Nearly all the charges are expelled by using a large capacitor as the sampling capacitor. Then, the readout voltage is obtained, and the difference is calculated to obtain the remanent polarization. Due to the use of a large capacitor, the voltage values during readout are relatively low, hence demanding a high circuit resolution. This approach requires accurate large capacitor values and high-precision voltage differences.

However, ferroelectric materials exhibit non-ideal effects, leading to unequal positive and negative remnant polarization of ferroelectric capacitors. The main factor responsible for this is the imprint effect. The imprinting effect of ferroelectric capacitors causes a shift in the hysteresis loop, as shown in Figure 7a. The forward movement of the hysteresis loop results in a decrease in the charges *Q*_0_ and *Q*_1_, disrupting the symmetry of the ferroelectric capacitor and leading to previous schemes. Moreover, the imprint effect significantly impacts the readout operation, causing insufficient readout and incomplete writing, as shown in Figure 7c,d. Therefore, the detection of the imprint effect is necessary.

This paper proposes a method of reading “0” to determine whether the ferroelectric capacitor is affected by the imprint effect and a method for detecting the offset voltage of the ferroelectric capacitor. Under symmetric conditions, the charge transferred when the ferroelectric capacitor polarizes from state *Q*_0_ to *Q*_0*sat*_ is equal to the charge transferred when it polarizes from state *Q*_1_ to *Q*_1*sat*_.

If the transferred charges for the two cases are unequal, it indicates an asymmetric hysteresis loop and the occurrence of the imprint effect. In this case, the offset amount needs to be detected. Additionally, Figure 7b simulates the increased charge transfer from *Q*_0_ to *Q*_0*sat*_ and the decreased charge transfer from *Q*_1_ to *Q*_1*sat*_ when the hysteresis loop is forwardly shifted. Due to the symmetric nature of the ferroelectric capacitor’s hysteresis loop, it can be inferred that by comparing the positive and negative directions, reading the data “0” can determine the direction of the imprint effect offset.


(2)
$$\begin{array}{cc} Q_{0sat\;}-Q_{0}=\left|Q_{1sat\;}-Q_{1}\right| & \Delta V_{shift\;}=0\\ Q_{0sat\;}-Q_{0}>\left|Q_{1sat\;}-Q_{1}\right| & \Delta V_{shift\;}>0\\ Q_{0sat\;}-Q_{0}<\left|Q_{1sat\;}-Q_{1}\right| & \Delta V_{shift\;}<0 \end{array}$$


The imprint effect can be considered equivalent to connecting a voltage source in series with the ferroelectric capacitor, resulting in voltage distortion applied across the two terminals of the ferroelectric capacitor. Therefore, by using a capacitor to compensate for the reverse voltage, the impact of the imprint effect can be counteracted, ensuring that the transferred charges for reading data “0” in the positive and negative directions are equal. The compensating voltage is the offset voltage value in this case.

Figure 8 illustrates the flip-readout scheme for compensating the imprint effect in ferroelectric capacitors. The flip-readout is achieved by switches S2–S5, and switches S1, S6–S10 are connected to different voltages for offset voltage compensation and readout operation. Capacitor C_SAM_ is the sampling capacitor used to receive the charges expelled from the ferroelectric capacitor, and capacitor C_ISO_ is the compensating capacitor used to compensate for the offset voltage. The specific scheme is shown in Table 2.

To perform the flip-readout operation, starting from *Q*_1_ to *Q*_1*sat*_, switches S1, S4, S6 and S9 are ON to precharge the ferroelectric capacitor and bring the voltage on the sampling capacitor to the ground. Then, polarization is performed, and the ferroelectric capacitor is polarized to *Q*_1*sat*_ using the signal V_PUL1_. After that, it is restored to the remanent polarization state *Q*_1_, and switch S6 is OFF, connecting the sampling capacitor only to the ferroelectric capacitor. The signal V_PUL1_ is pulled high, and the charges on the ferroelectric capacitor are transferred to the sampling capacitor. The specific transfer charge can be expressed using the formula in the table, representing the intersection of the red load line and the hysteresis loop.

Similarly, the voltage generated across the sampling capacitor reflects the charge transferred from *Q*_0_ to *Q*_0*sat*_ during the readout process. If the readout voltage of *Q*_0_ is higher, it indicates a positive offset voltage. An example will be provided using a positive offset.

In order to compensate for this positive offset, a compensation voltage is constructed. Firstly, the process of *Q*_1_ to *Q*_1*sat*_ is initiated. The switches S1, S3, S4, S6, S8, and S10 are ON. Precharging is performed by grounding V_PUL1_, clearing the charge on both ends of the isolation capacitor, and applying the compensating bias voltage V_IM_ to V_PUL2_ and V_PUL4_ to ensure that the voltage difference between C_SAM_ and C_ISO_ is 0. Then, S1 is OFF and S7 is ON. V_PUL3_ is connected to the electrode of C_ISO_ and a saturation voltage pulse is applied. Due to the voltage difference being 0 across C_ISO_, the negative electrode of the ferroelectric capacitor will be at saturation voltage, while the positive electrode voltage will be V_IM_, thus writing the ferroelectric capacitor into a polarized state after compensation. After the pulse, it remains in the compensated remnant polarization state. Then, S6 is OFF and V_PUL3_ applies the readout voltage. The accumulated charge obtained through C_SAM_ represents the value after correcting for the V_IM_ bias voltage. During the readout, S9 is ON and S10 is OFF, thereby eliminating the influence of the offset voltage. Similarly, the amount of charge transferred from *Q*_0_ to *Q*_0*sat*_ after compensation is read. The voltage is not applied exactly at once; it is increased gradually by varying the V_IM_ voltage. If the readout voltages are unequal, the offset voltage value continues to increase until they become equal. In this point, V_IM_ represents the correct offset voltage.

Once the imprint voltage is obtained, the readout voltage can be corrected, as shown in Table 3, to obtain the remanent polarization and saturation polarization values without the imprint effect. Taking negative offset as an example, the first step is to precharge the offset voltage on both sides of the C_SAM_ capacitor. Then, V_PUL2_ is pulled high to expel the charge in C_FE_, and S9 is OFF while S10 is ON to pull the lower electrode of the sampling capacitor low. The transferred charge solely generates the remaining voltage values in C_SAM_.

## 3. Specific Circuit Design and Simulation Results

Figure 9 shows the overall circuit architecture and design of a capacitive readout circuit for ferroelectric capacitors. The circuit, implemented in a 130 nm process, consists of a 16 Kb array with a configuration of 128 × 128. To minimize the impact of parasitic capacitance on the sampling capacitor, a reduced number of bitline loads is chosen. The array is connected to the capacitor array through column select switches and flip-flop switches. Both the column decoder circuit and the row decode circuit are constructed using 3-to-8 decoders and 4-to-16 decoders. The driving is composed of an inverter chain.

A readout circuit reads out the voltage on the capacitors. The readout circuit includes a comparator, a current source, and a 6 bit digital time-to-digital converter (TDC). The peripheral circuitry includes a voltage generation circuit, decoder, timing generation circuit, capacitor, and a TDC calibration circuit. The entirety of the digital control is assembled using digital components from the standard library. Timing is achieved based on a 200 MHz clock, and the voltage reference employs a basic voltage reference module, with external lead routing also implemented to prevent reference source errors.

The signal path and flip-read control switch are depicted in Figure 9b. The circuit employs a 1T1C cell structure, with the column select circuit utilizing a two-layer 4-to-1 MUX, combined with a single-layer 8-to-1 MUX. Flip switches are all implemented using 4 times the minimum size transmission gates to ensure both the speed and accuracy of signal transmission. The capacitor CISO, being an isolation capacitor, does not require high precision, and hence is designed to be 0.8 pF.

Figure 10 shows the specific timing diagram and operation process of the ferroelectric capacitor testing circuit. The enable signal of the chip triggers the updates of V_PUL1-4_ and S1–S11 signals. After the address signal is ready, the wordline (WL) is opened, and then the imprint effect compensation and correct polarization value writing of the ferroelectric capacitor begin. If no compensation is needed, this period is idle. When switches S9 and S11 are ON and switches S2, S4, and S10 are OFF, the TDC enable signal is high, and the sensing readout starts. After obtaining the quantization value, a reset is performed for all nodes, and then it waits for the subsequent readout.

As analyzed in the previous text, since the measurement is of charge quantity and requires the charge to be completely transferred to the sampling capacitor, accurate capacitance values are needed, and the capacitance value is relatively large. Therefore, a variable capacitor is designed and a capacitance calibration circuit is included. In this paper, simulations were performed on a ferroelectric capacitor with an area of 0.7 × 0.7 μm^2^. The charge transferred when the ferroelectric capacitor is polarized from *Q*_1_ to *Q*_0*sat*_ and from *Q*_0_ to *Q*_0*sat*_ was investigated under different capacitance values of the sampling capacitor. The simulation result is demonstrated in Figure 11a. It was noted that when the capacitance is 1 pF, the transferred charge has already reached saturation. Accordingly, we adopt a capacitance value of 1 pF to save area.

However, there is a 15% deviation caused by capacitor manufacturing. To compensate for this deviation, a capacitance calibration circuit was designed, as shown in Figure 11b. The circuit includes a comparator, capacitor array, register group, 3 μA current source, and control circuit. The capacitance array consists of one 800 fF capacitor, four 100 fF capacitors, and four 20 fF capacitors (due to process limitations, the minimum capacitor value is 16 fF, so 20 fF was chosen). The combinable range of capacitance values is from 800 fF to 1180 fF, with a minimum spacing of 20 fF. Considering the 15% deviation in the capacitor manufacturing process, the worst-case scenario is that the manufactured value is 115% and 85% of the design value. For the designed capacitance array, the minimum value of 800 fF × 115% is less than 1000 fF, and the maximum value of 1280 fF × 85% is larger than 1000 fF, so the entire range is covered.

The calibration process is shown in Figure 12. First, the 800 fF capacitance and two 100 fF capacitances are connected. Then, when the calibration enables signal is high, a 3 uA current source charges the capacitor for 100 ns on the rising edge of a 5 MHz clock. Afterward, the voltage is compared with 300 mV by a comparator. Suppose the readout voltage is greater than 300 mV and the comparison result is smaller. In this case, another 100 fF capacitor is connected, and the comparison is repeated until the output of the comparator can be written by a 20 fF capacitor. Therefore, the offset voltage of the comparator needs to be able to distinguish a capacitance value of 20 fF.

The current source adopts a common source common gate structure, and the highest output voltage is 300 mV. It has tiny fluctuations across the entire process corner, so it can be considered constant as shown in Figure 13. The time an external clock generates T, and the offset can be neglected, so I*× is a constant. The minimum resolution to be distinguished is when the capacitor is at the minimum deviation, ΔC = 20 fF × 0.85 = 17 fF. Therefore, the offset voltage range is $V_{\mathrm{os}}<\mathrm{Min}\left\{\left(\frac{I\times T}{1\;\mathrm{pF}-17\;\mathrm{fF}}-\frac{I\times T}{1\;\mathrm{pF}}\right),\left(\frac{I\times T}{1\;\mathrm{pF}}-\frac{I\times T}{1\;\mathrm{pF}+17\;\mathrm{fF}}\right)\right\}=5.01\;\mathrm{mV}$. Therefore, the offset voltage of the comparator is designed to be 5 mV. The specific circuit implementation of the comparator is shown in Figure 14a. The biggest source of the comparator’s offset voltage comes from the threshold voltage deviation.
(3)$$\begin{array}{c} V_{\mathrm{offert}\;}=\sqrt{\Delta {\mathrm{Vth}}_{12}{}^{2}+\frac{{\mathrm{gm}}_{3}{}^{2}}{{\mathrm{gm}}_{1}{}^{2}}\Delta {\mathrm{Vth}}_{34}{}^{2}+\frac{{\mathrm{gm}}_{5}{}^{2}}{{\mathrm{gm}}_{1}{}^{2}}\Delta {\mathrm{Vth}}_{56}{}^{2}},\;\Delta \mathrm{Vth}=\frac{A_{\mathrm{VTH}}}{\sqrt{\mathrm{WL}}} \end{array}$$

To reduce the comparator’s offset voltage, the sizes and current flow of transistors M1 and M2 are increased to increase gm1. At the same time, the areas of transistors M1 and M2 are increased to reduce the influence of threshold voltage, and the source and substrate are connected to reduce the influence of substrate bias voltage. Monte Carlo simulation was performed with 1000 data points, with a mean of 37.72 μV and a standard deviation of 1.39 mV, shown in Figure 14b. The offset voltage is less than 5 mV within 3 times σ, meeting the design requirements.

After calibration, the maximum deviation of the 1 pF capacitor is 49 fF (affected by the comparator’s offset voltage and process corner), which is 84% less than the deviation of 300 fF as shown in Figure 15.

After completing the calibration process, the charge on the capacitor can be accurately measured. For the target ferroelectric capacitor area of 0.7 μm × 0.7 μm, the difference between the negative remaining polarization charge *Q*_1_ and the positive saturation polarization charge *Q*_0*sat*_ is 290 fC. Figure 11a simulates the charge transferred to the capacitor from *Q*_0*sat*_ to *Q*_1_ using a 1 pF sampling capacitor, which amounts to 281 fC. This indicates that the dynamic range requirement must exceed 281 fC. Furthermore, the charge transferred from *Q*_0*sat*_ to *Q*_0_ is 48 fC. In order to achieve an accuracy of approximately 10%, a minimum resolution of 5 fC is designed.

Figure 16 illustrates the specific readout circuit, including the variable current source, variable capacitor, and comparator. The incorporation of a variable current source aims to facilitate the calibration of the TDC circuit, thereby achieving enhanced precision. The collective functionality of these three components enables the conversion of voltage signals into delay signals (voltage-to-time conversion circuit VTC). The simplified process involves charging the capacitance using the current source. When the charging reaches the reference voltage, the comparator flips, and the delay from the beginning of charging by the current source to the high state of the comparator is recorded. Therefore, the reference voltage needs to be set higher than the maximum voltage after the capacitance collects the charge. With 281 fC accumulated in a 1 pF capacitor, a voltage of 281 mV is generated. Hence, Vref2 is designed as 300 mV, and the half-range reference Vref1 is set to 150 mV.

The signal is then passed to a 6 bit subtraction TDC for quantized readout. The TDC circuit consists of SR triggers, a half-range delay cell (HRDC), 32 TDC slices, and 31 XOR gates. The HRDC improves precision a bit. The quantization process involves the detection of begin and end signals by the SR triggers, with the XOR gates detecting which slice receives the end signal when the end signal arrives.

The TDC slice cell is designed as shown in Figure 17, using a current-starvation-based delay cell and D flip-flops. In the readout operation, all outputs transition from low to high. To ensure prompt sampling of the terminating signal, the “end” signal is concurrently linked to both the NMOS input of the second stage and the clock input of the trigger. The current flowing through the unit is controlled by an external current source, ensuring consistent maximum current flow through each unit during operation. Compared to voltage-controlled delay units, this design exhibits better tolerance to process variations.

The structure of the HRDC is depicted in Figure 18, consisting of six cascaded voltage-controlled current starvation delay units. As it functions as an overall delay module, a large adjustment range is required. Hence, voltage control is chosen.

The overall timing diagram of the read circuit operation is illustrated in Figure 19a, while Figure 19 depicts specific simulation instances of signals “begin,” “end,” “Z,” and “D.” The simulation is initialized with an initial voltage of 230 mV and a current source of 1.25 μA. The simulation results show that D15 is high, aligning with the predefined objective. The detailed principal analysis is as follows:

When TDC_EN is enabled and S_ref is low, Vref1 is connected. First, Vcsam is compared to 150 mV, and if it is greater, the charging time to reach 300 mV is less than half the range, and S_ref becomes high, shorting HRDC. If V_CSAM_ is less than 150 mV, the charging time to reach 300 mV exceeds half the range, and S_ref remains low, connecting HRDC to the circuit to expand the range. Once the S_ref signal is locked, Read_EN is high, and through RS triggers, the begin signal is high while the end signal is low. At this point, both the current source and TDC slides start working. The C_SAM_ capacitor is charged, Vcsam approaches 300 mV, and Z0–Z30 is sequentially high. When Vcsam exceeds 300 mV, the comparator outputs high, resetting the RS trigger. The end signal becomes high, and the begin signal becomes low. At this point, the propagation of the Z signal is interrupted, and simultaneously, the Z signal is latched by the trigger. The propagation location is determined using XOR gates, resulting in a high output. This is depicted in the diagram where D15 is in the high state. Then, the data are converted into binary format through decoding circuits. Afterward, TDC is reset.

Using a time-to-digital converter (TDC) circuit as the readout circuit offers several advantages. First, it is easy to implement and has a simple circuit design. Second, it provides fast readout speed. Third, as the readout path is unique, the TDC circuit is suitable for differential measurements, eliminating offset voltages along the path. Fourth, it can be reused with capacitance calibration circuits.

However, process variations can lead to significant delays in inverter delay cells, even exceeding 40%, resulting in large deviations between the designed initial values and the actual values. Calibration is required to compensate for these discrepancies. The delay modules in the circuit consist of TDC slices and HRDC, both of which require calibration.

First, the calibration of TDC slides is performed as shown in Figure 20a. Initial values, such as I_delay_ and I_cap_, are set. Then, the entire TDC is reset, C_SAM_ voltage is set to 0, and Vref is set to 150 mV. The TDC starts working, and I_cap_ begins charging the capacitor. When the charging reaches 150 mV, the TDC stops working, and D29 is checked. If it is low, Q29 is checked. If Q29 is high, it indicates that the capacitor is charging quickly, so I_cap_ is reduced. If Q29 is low, I_cap_ is increased until D29 becomes high. At this point, the calibration is completed. Figure 20b depicts the TDC slide calibration flowchart. The resulting equation can be expressed as follows:
(4)$$\begin{array}{c} \frac{\;\mathrm{Vref}\;\times \;C_{\mathrm{sam}\;}}{I_{\mathrm{cap}}}+\mathrm{tos}=\sum_{0}^{29}\mathrm{td} \end{array}$$

The left side of the equation represents the delay and time offset, tos, of I_cap_ charging the capacitor C_SAM_ to Vref. The right side represents the delay through 30 TDC slices. The value of td, representing the delay unit of each TDC slice, is a function of I_delay_, while tos represents the propagation delay of the comparator. The deviation of td between units can be kept within 1% by adding capacitors after the delay units [20]. Therefore, this paper ignores the influence of td deviations between units. The propagation delay of the comparator can be reduced by increasing the current, adjusting the gain, and setting the common-mode level properly.

The simulation in Figure 21 demonstrates that the average delay of the comparator and subsequent driving is 2.02 ns with a standard deviation of 0.505 ns. This delay has a relatively minor impact on the overall propagation and can be approximated as negligible. Therefore, the final precision is achieved.
(5)$$\begin{array}{c} \;\mathrm{Vref}\;\times C_{\mathrm{sam}}=30\times \mathrm{td}\times I_{\mathrm{cap}}\\ \mathrm{td}\left(I_{\mathrm{delay}}\right)\times I_{\mathrm{cap}}=5\mathrm{fC} \end{array}$$

The calibration process for the HRDC module is shown in Figure 22a. Initially, the values of I_delay_, I_cap_, and Vcon are set. The TDC is reset, and Csam is charged. When the charging reaches 150 mV, D0 is checked. If it is not high, *Q*_0_ is checked. If *Q*_0_ is high, Vcon needs to be reduced to increase HRDC delay. If *Q*_0_ is low, Vcon is increased to decrease HRDC delay. This process continues until D0 becomes high. At this point, HRDC delay corresponds to 150 mV. This doubles the range of the TDC. Figure 22b depicts the HRDC calibration flowchart.

Figure 23a illustrates the ENOB testing process. Firstly, a sine signal generator generates a sinusoidal wave with a center frequency of 500 kHz and a voltage amplitude ranging from 0 to 300 mV. This waveform is then input to the circuit under test. For the output results, 2048 points are sampled at a frequency of 5 MHz. The data are collected in a CSV file and subsequently imported into MATLAB for FFT analysis to obtain an effective number of bits (ENOB) of 5.91.

Due to the output being 31 bits instead of 32 bits, the inherent number of bits is 5.95. Subsequently, owing to the propagation delay of the comparator, inherent inaccuracies in the capacitor calibration circuit, and the influence of bit-line parasitic capacitance, a certain level of precision is lost. As a result, the effective number of bits is ultimately 5.91 bits.

## 4. Conclusions

This paper provides a detailed analysis of the readout process of ferroelectric capacitors and based on this analysis, proposes a scheme for testing the remanent polarization, saturation polarization, and imprint effect of small-area ferroelectric capacitors. Furthermore, specific circuit designs are presented, including a flip-readout circuit, a capacitance calibration circuit, and a VTC&TDC (voltage-to-time converter and time-to-digital converter) readout circuit.

The flip-readout circuit is used for detecting and compensating for imprint effects, while the capacitance calibration circuit achieves high-precision capacitance measurements, reducing the capacitance error by 84% for a 1 pF capacitor. The VTC&TDC circuit achieves a precision of 5.91 bits, with a design precision of 5 fC, a readout time of 150 ns, and a power consumption of 1 mW.

Summarizing the technology presented in this paper once again, the data to be extracted in this paper include imprint offset voltage, saturated polarization under no imprint effect, and remnant polarization under no imprint.

The following flowchart presents the overall design scheme, along with a detailed flowchart illustrating the measurement of imprint offset voltage is shown in Figure 24.

The Comparison with Prior Work is shown in Table 4. The read circuit sensing time only represents the direct readout units, excluding compensation and polarization processes.

The power consumption is measured during the readout time of 150 ns by assessing the current flowing through VDD. This includes the current of the delay unit, comparator current, and auxiliary currents in the peripheral circuit such as the reference source, triggers, and digital circuitry. The absolute values of these currents are taken, and the average is calculated.

The delay refers to the time from when the Read En signal goes high to the reading out of data, which is 150 ns.

Compared to the work in the JSSC paper from 2013, this paper demonstrates faster readout speed, higher effective bit resolution, and the capability to compensate for process deviations through calibration. The technology presented in this paper exhibits faster readout speeds and acquires data in a single sampling step, whereas the JSSC technology requires two sampling steps and reads out data at a slightly slower rate. The precision of this paper is improved by utilizing half-range extension technology to achieve a higher number of effective bits. This design can calibrate delay current and delay voltage, compensating for temporal losses due to process deviations and achieving a testing accuracy of approximately 5 fC.

Compared to the TCAS-II work from 2013, the approach in this paper achieves a higher bit resolution, lower power consumption, and reduced design complexity. The TCAS-II technology employs a 4 bit Flash ADC with high power consumption per single readout and requires the design of 16 comparators, occupying a larger area and posing greater design challenges. The technology presented in this paper utilizes inverter and flip-flop designs, resulting in simpler unit cells and lower design complexity. Furthermore, this technology analyzes the impact of process variations on accuracy and provides corresponding calibration strategies.

This paper introduces a testing circuit that offers a cost-effective, high-speed, and precise solution for testing ferroelectric capacitors. In the future evolution of ferroelectric arrays, this circuit could be seamlessly integrated, eliminating the need for previously utilized off-chip testing instruments characterized by lower accuracy, slower speeds, and higher expenses. This integration would accelerate development cycles and propel advancements in array-level ferroelectric capacitor technologies.

Moreover, this circuit can serve as an adept readout solution for ferroelectric capacitors, offering high precision, capacity, and compactness. It holds potential for applications in circuits demanding direct detection of the residual polarization voltage of ferroelectric capacitors. Additionally, this technology can be extended to multi-level ferroelectric capacitor memory systems.

The circuit designed in this paper has the potential for application in advanced technology node. The primary challenges in advanced technology node are significant process variation and low operating voltage. The capacitance calibration circuit and the TDC calibration circuit designed in this paper can compensate for the impact of process variation. Additionally, most of the signal used in this paper is digital, allowing operation at low voltage. Moreover, comparator can also be implemented using low-voltage structure. Therefore, in the future, when hafnium-based ferroelectric capacitor is applied to even more advanced technology node, the test circuit designed in this paper can still be applicable.

This lays a solid foundation for the optimization of devices and the design of circuits in hafnium-based ferroelectric capacitor memories.

## Figures and Tables

**Figure 1 micromachines-14-01851-f001:**
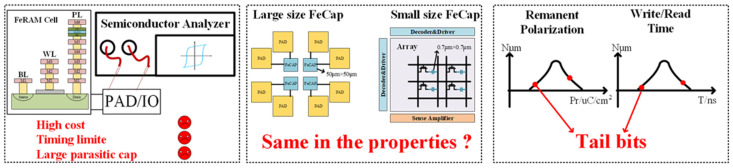
Challenges faced by small-area ferroelectric capacitors.

**Figure 2 micromachines-14-01851-f002:**
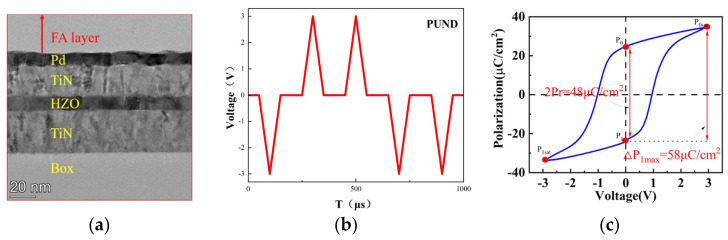
(**a**) TEM cross-section of a 50 μm × 50 μm Hf_0.5_Zr_0.5_O_2_ (HZO) FeCAP; (**b**) the scanning waveform applied during testing; (**c**) the measured hysteresis loop of the capacitor.

**Figure 3 micromachines-14-01851-f003:**
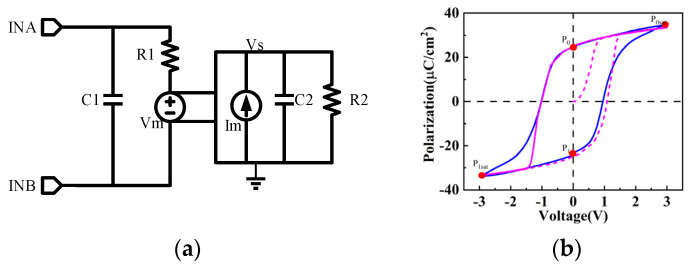
(**a**) The circuit-level switch model of the ferroelectric capacitor; (**b**) the simulation fitting of the ferroelectric capacitor model.

**Figure 4 micromachines-14-01851-f004:**
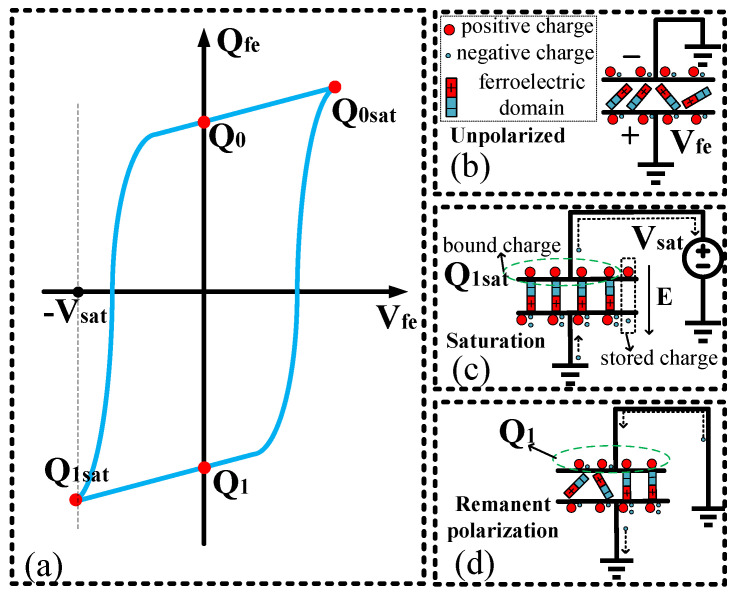
The writing process of a ferroelectric capacitor; (**a**) Hysteresis loop for the ferroelectric capacitor: *Q*_0_ and *Q*_1_ are remanent polarization, while *Q*_0*sat*_ and *Q*_1*sat*_ are saturated polarization; (**b**) Initial state of the ferroelectric capacitor; (**c**) Ferroelectric capacitor in *Q*_1*sat*_ saturated polarization; (**d**) Ferroelectric capacitor with remanent polarization *Q*_1_.

**Figure 5 micromachines-14-01851-f005:**
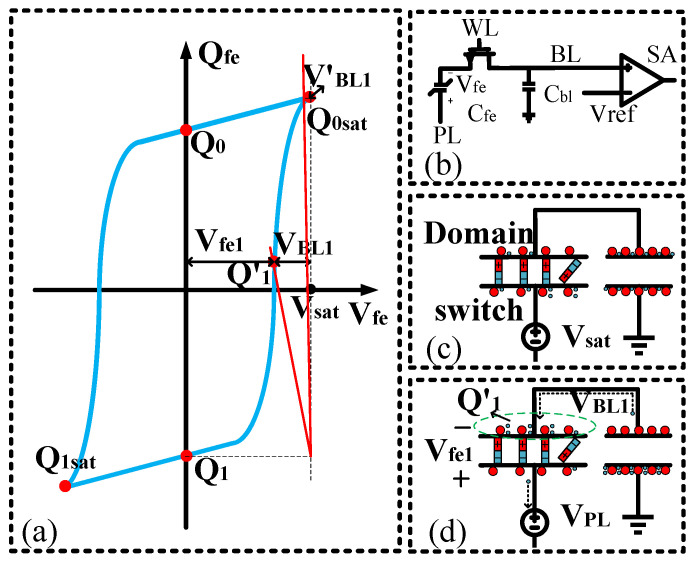
The read process of a ferroelectric capacitor. (**a**) Different bitline capacitance readouts yield distinct readout voltages, *V_BL_*_1_ and *V’_BL_*_1_; (**b**) Ferroelectric capacitor readout circuit; (**c**) Simplified circuit for ferroelectric capacitor readout; (**d**) Ferroelectric capacitor readout involving electron transfer.

**Figure 6 micromachines-14-01851-f006:**
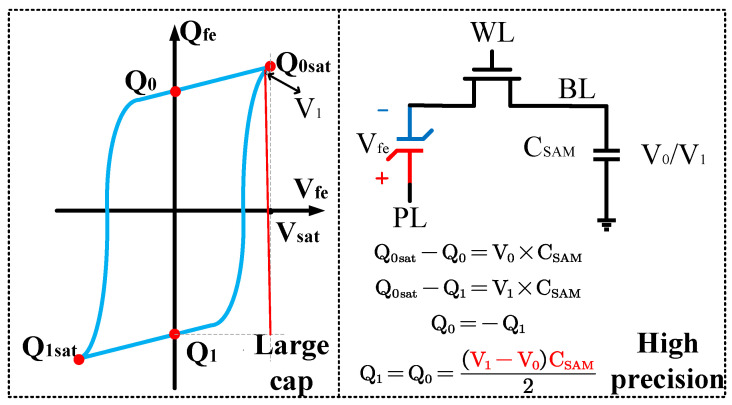
Measurement of remnant polarization.

**Figure 7 micromachines-14-01851-f007:**
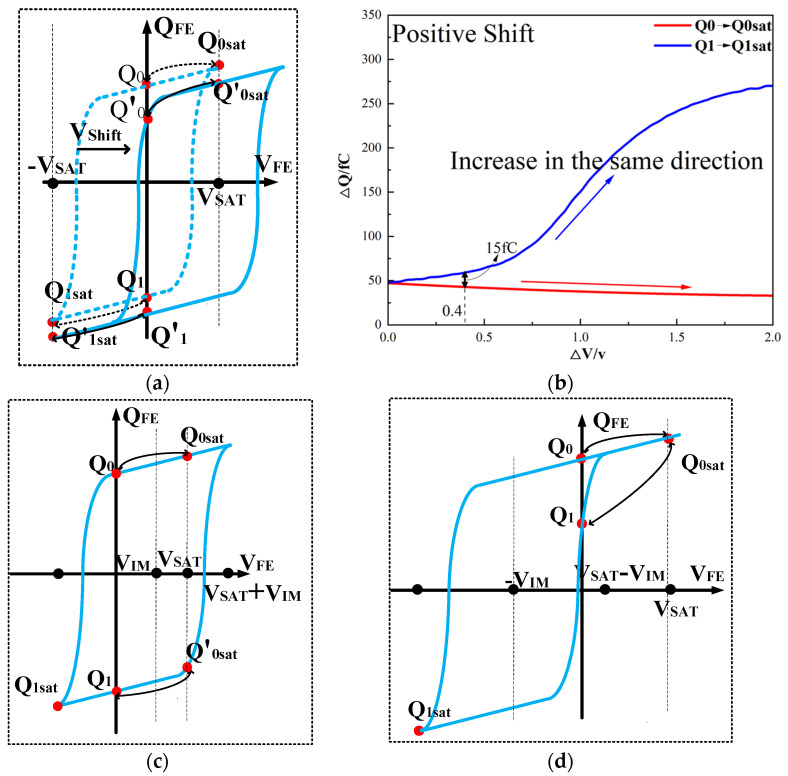
(**a**) Impact of imprint on hysteresis loop; (**b**) positive drift of the hysteresis loop, charge transfer for reading “0”; (**c**) reading “0” and “1” during positive biasing; (**d**) reading “0” and “1” during negative biasing.

**Figure 8 micromachines-14-01851-f008:**
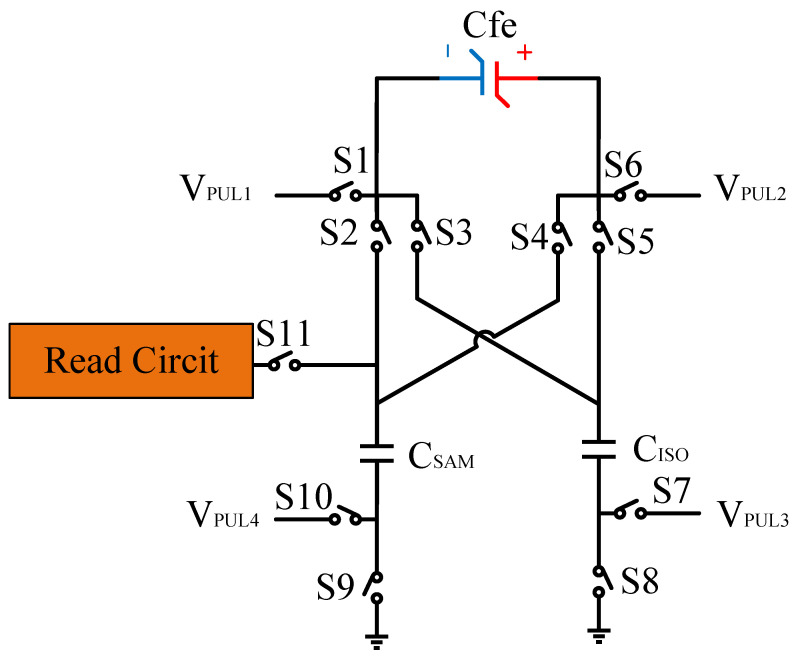
The flip-readout scheme for compensating the imprint effect in ferroelectric capacitors.

**Figure 9 micromachines-14-01851-f009:**
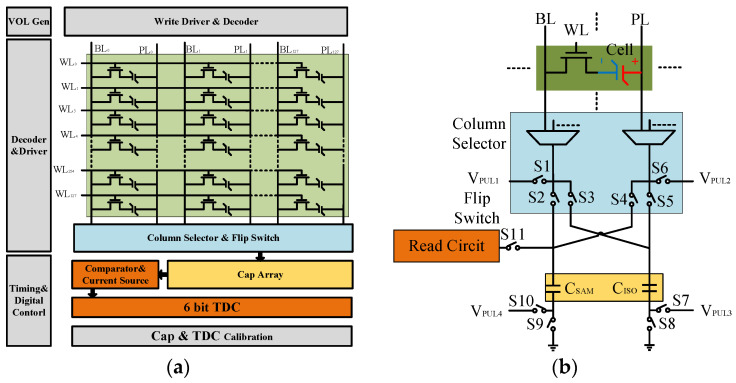
(**a**) Overall architecture of the ferroelectric capacitor test chip; (**b**) specific measurement path during testing.

**Figure 10 micromachines-14-01851-f010:**
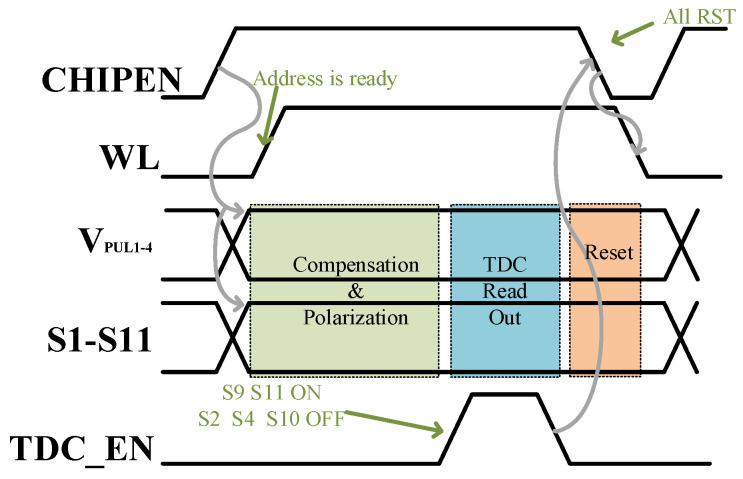
Timing diagram of the ferroelectric capacitor test circuit operation.

**Figure 11 micromachines-14-01851-f011:**
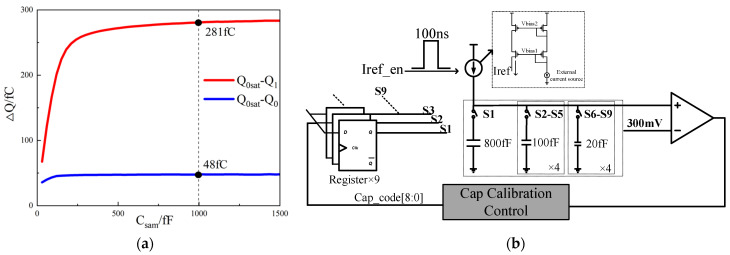
(**a**) The charge accumulation in the normal case when *Q*_1_ is polarized to *Q*_0*sat*_ and when *Q*_0_ is polarized to *Q*_0*sat*_ under different sampling capacitor values; (**b**) the capacitance calibration circuit.

**Figure 12 micromachines-14-01851-f012:**
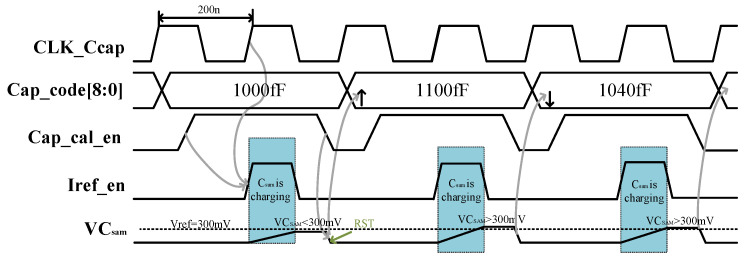
Capacitor calibration circuit operation timing.

**Figure 13 micromachines-14-01851-f013:**
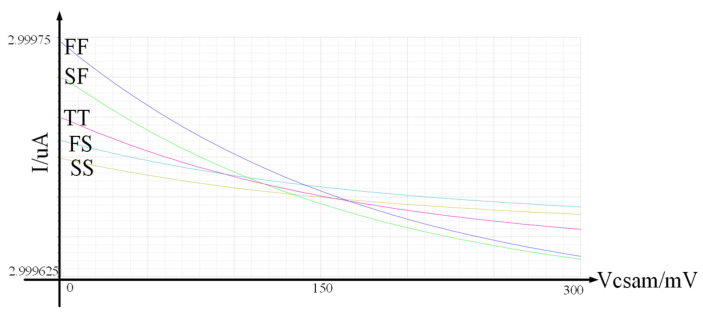
Under different process corners, the output current varies as the load voltage changes from 0 to 300 mV.

**Figure 14 micromachines-14-01851-f014:**
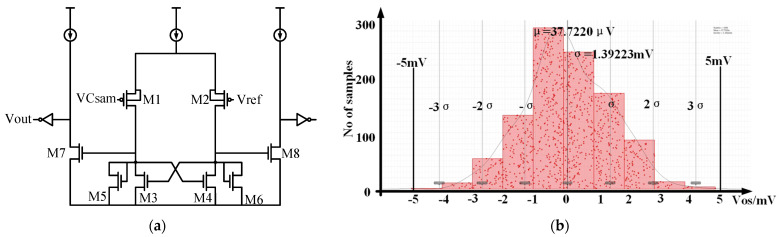
(**a**) The specific circuit implementation of the comparator; (**b**) the Monte Carlo simulation of the comparator’s offset voltage.

**Figure 15 micromachines-14-01851-f015:**
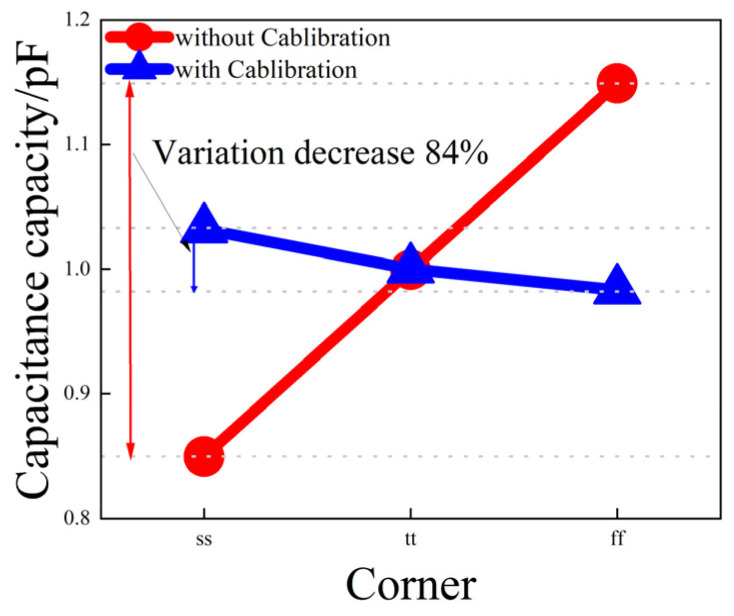
The variation of the 1 pF MIM capacitor value with process corner, both with and without calibration.

**Figure 16 micromachines-14-01851-f016:**
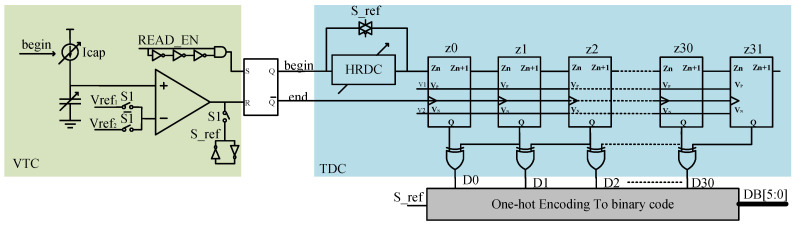
The specific readout circuit.

**Figure 17 micromachines-14-01851-f017:**
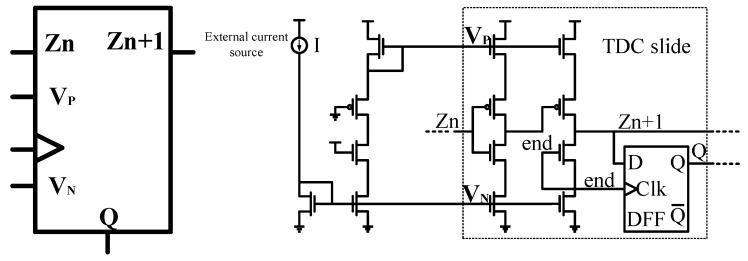
TDC slide circuit.

**Figure 18 micromachines-14-01851-f018:**
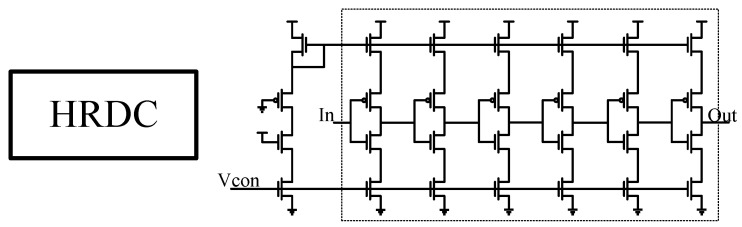
HRDC delay module circuit design.

**Figure 19 micromachines-14-01851-f019:**
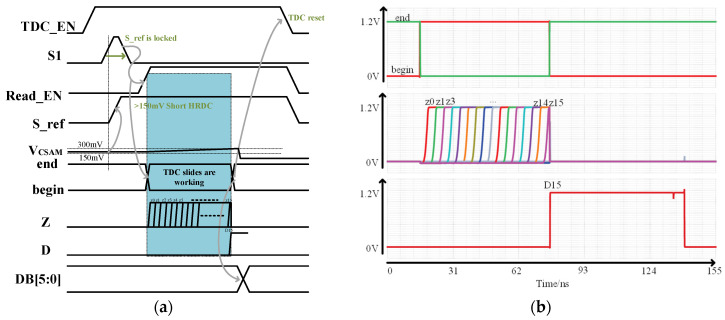
(**a**) Read circuit operation timing; (**b**) simulation timing diagram of the readout circuit.

**Figure 20 micromachines-14-01851-f020:**
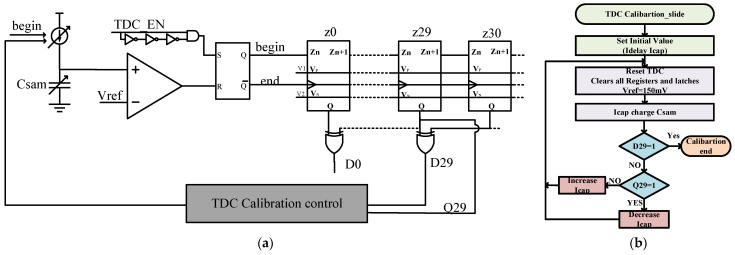
(**a**) TDC slide calibration circuit; (**b**) TDC slide calibration flowchart.

**Figure 21 micromachines-14-01851-f021:**
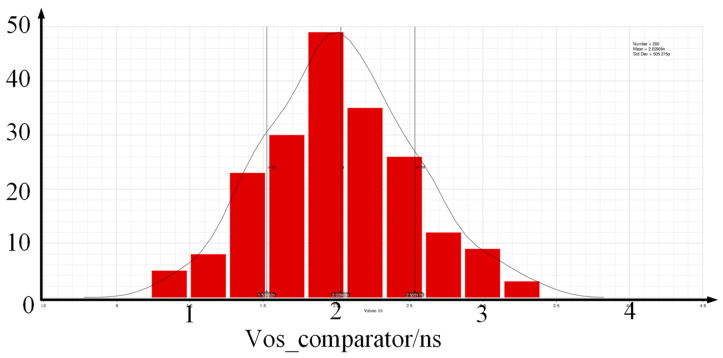
Comparator propagation delay Monte Carlo simulation.

**Figure 22 micromachines-14-01851-f022:**
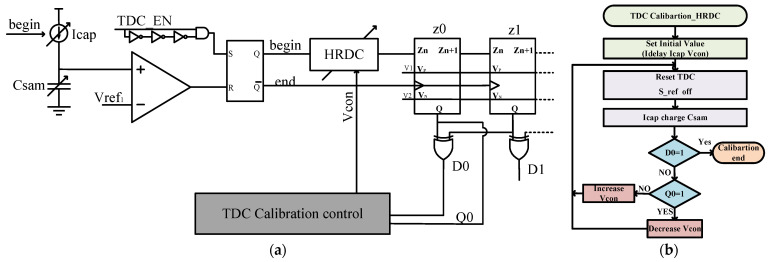
(**a**) HRDC module calibration circuit diagram; (**b**) HRDC module calibration flowchart.

**Figure 23 micromachines-14-01851-f023:**
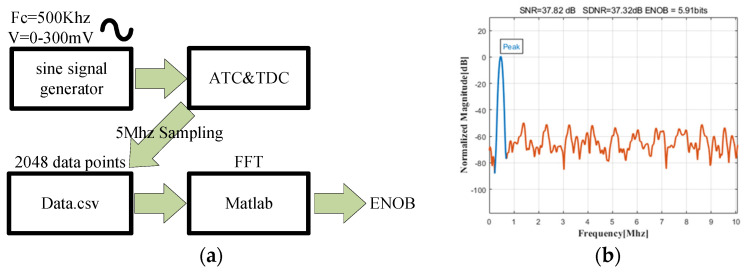
(**a**) Process of effective number of bits (ENOB) Testing; (**b**) ENOB of the readout circuit.

**Figure 24 micromachines-14-01851-f024:**
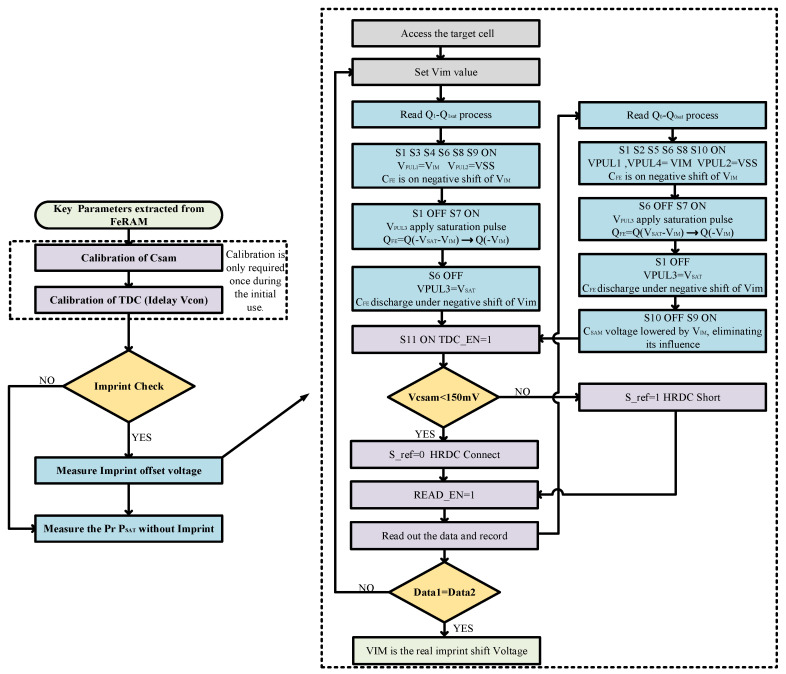
Testing process for small-area ferroelectric capacitors and detailed procedure for measuring imprint offset voltage.

**Table 1 micromachines-14-01851-t001:** Specific Parameters of Figure 3a.

Model Parameter	Value
A	−5.2245 × 10^12^
B	−2.2423 × 10^38^
C	4.1090 × 10^64^
C1	1 fF
C2	1 F
R1	1 MΩ
R2	1 GΩ

**Table 2 micromachines-14-01851-t002:** Flip-Readout and Imprint Offset Compensation.

State 1	Process	Precharge	Polarization	Read Out
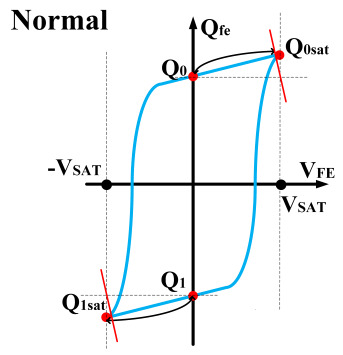	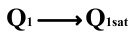	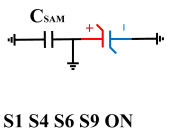	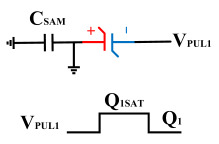	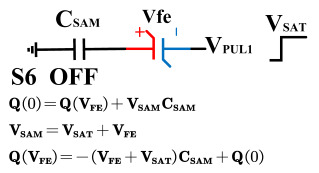
	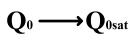	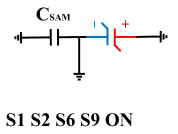	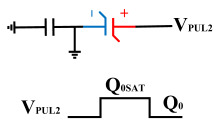	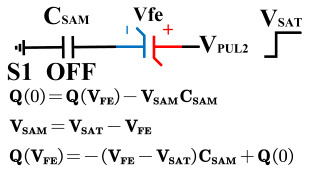
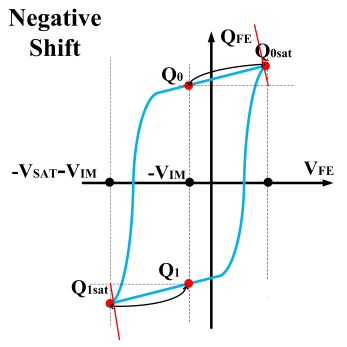	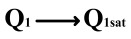	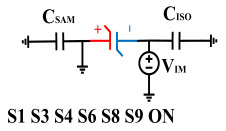	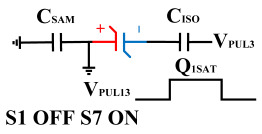	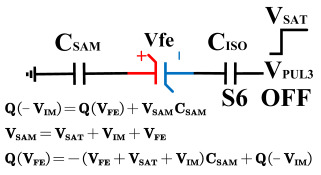
	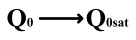	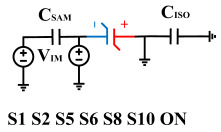	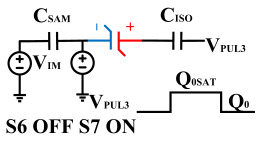	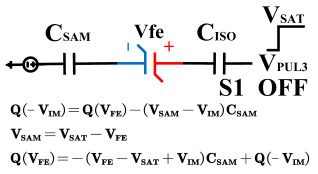
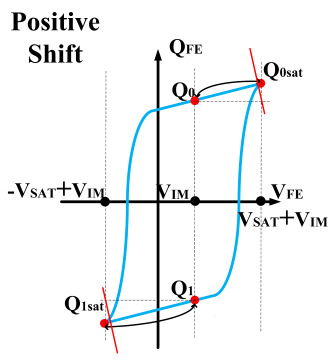	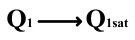	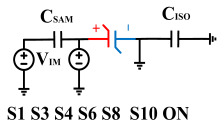	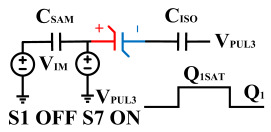	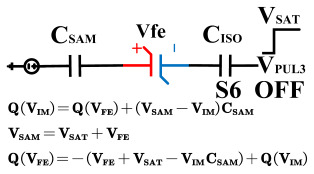
	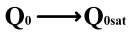	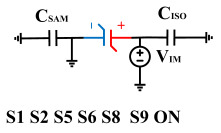	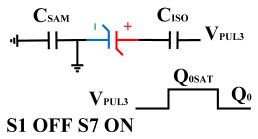	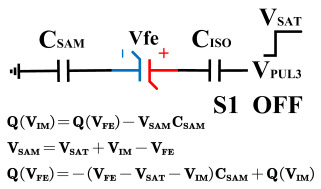

**Table 3 micromachines-14-01851-t003:** Readout of Remanent Polarization Charge.

No Imprint	Imprint-Positive	Imprint-Negative
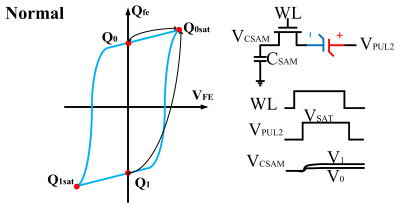	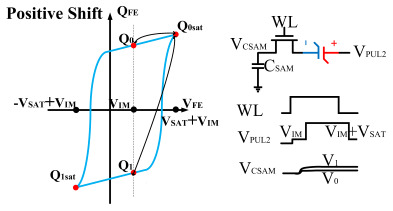	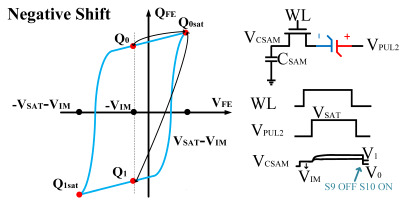

**Table 4 micromachines-14-01851-t004:** Comparison with Prior Work.

	This Work	JSSC 2013 [21]	TCAS-II 2013 [22]
Technology	130 nm	130 nm	65 nm
Application	FeRAM	FeRAM	RRAM
Read circuit	VTC&6 bit TDC	VTC&5 bit TDC	4 bit Flash ADC
adaptive range	YES	NO	NO
Calibration	YES	NO	NO
Read Circuit Sense Time	150 n	200 n	10 n
Read Circuit Power	1 mW	0.8 mW	15 mW

## Data Availability

The data that support the findings of this study are available from the corresponding author upon request.

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
