# Peer review of "Methodology for Testing Key Parameters of Array-Level Small-Area Hafnium-Based Ferroelectric Capacitors Using Time-to-Digital Converter and Capacitance Calibration Circuits"

_micromachines, 2023, doi:10.3390/mi14101851_

Round 1

Reviewer 1 Report

1. The proposed test circuit overcomes these limitations and enables high-precision testing of ferroelectric capacitors, contributing to the development of haf-nium-based ferroelectric memories. The circuit includes a flip-readout circuit, a capacitance calibration circuit, and a Voltage-to-Time Converter and Time-to-Digital Converter (VTC&TDC) readout circuit.

2. In the figure 2, (a) TEM cross-section of a 50 μm × 50 μm Hf0.5Zr0.5O2 (HZO) FeCAP; (b) the measured hysteresis loop of the capacitor, should be elaborated in detail.

3. In the figure 8, (a) overall architecture of the ferroelectric capacitor test chip; (b) specific measurement path during testing, should be elaborated in detail.

4. In the figure 20, effective number of bits (ENOB) of the readout circuit, should be elaborated in detail.

5.Please compare the contributions of the proposed technology to related technologies, in detail.

6.Please discuss some of the future applications of the proposed technology, in detail.

7.Please thoroughly revise the language before your final submission.

Moderate editing of English language required.

Author Response

We would greatly appreciate it if you can review our paper in a timely manner. Thank you!

Reviewer 2 Report

This paper presented a methodology for testing key parameters of array-level ferroelectric capacitors. Here are my comments. 

1. Writing and figure quality need to be improved. It is also highly recommended to take English correction by native speakers. 

2. What is the ferroelectric switching model simulated in this work? 

3. The authors provided many circuits, but I couldn't find some relevant simulation results to verify the functionalities of the circuits. Please provide the results. 

4. How was the performance estimated? (i.e., power, latency, etc.) 

5. Please provide a summary that clearly states the key parameters to be extracted and how the parameters could be extracted by the circuits in detail (at least, please provide some examples). 

It is very hard to follow the manuscript and its claims. It is highly recommended to take English corrections. 

Author Response

(The authors gave the same response as above.)

Reviewer 3 Report

The proposed contribution addresses high-precision testing of ferroelectric capacitors.

The paper sounds as Hafnium-Based ferroelectric capacitors are under deep investigation.

A specialized high-precision test circuit for on-chip testing of small-area hafnium-based ferroelectric capacitors is presented to obtain the ferroelectric capacitor key parameters (such as remnant polarization, saturation polarization, and coercive field offset).

A 16Kb 128x128 FeRAM memory array circuit is designed and implemented in a 130nm process technology.

The circuit is validated through simulations (including monte-carlo simulations).

The capacitance calibration circuit improve the state-of-the-art.

Feedbacks to handle:

-       The choice of a more advanced technology node would bring more impact to the study.

-       Authors should elaborate on the use of a ferroelectric capacitor model at the simulation level (simple capacitor, dedicated model, etc.). 

Table 3 goes beyond the page limits

Fig. 1 quality should be improved

Fig. 6 quality should be improved. If extracted from another contribution, please indicate the reference

Author Response

Dear Reviewer:

Thank you very much for your attention and the referee’s evaluation and comments on our paper “Methodology for Testing Key Parameters of Array-Level Small-Area Hafnium-Based Ferroelectric Capacitors Using Time-to-Digital Converter and Capacitance Calibration Circuit”. We have revised the manuscript according to your kind advices and referee’s detailed suggestions. Enclosed please find the responses to the referees. We sincerely hope this manuscript will be finally acceptable to be published on Micromachines. Thank you very much for all your help and looking forward to hearing from you soon.

Best regards

Sincerely yours

Jianguo Yang

Round 2

Reviewer 1 Report

The authors have carefully revised the manuscript. I suggest to accept this manuscript.

Author Response

Dear Reviewer,

Thank you very much for your attention and the referee’s evaluation and comments on our paper “Methodology for Testing Key Parameters of Array-Level Small-Area Hafnium-Based Ferroelectric Capacitors Using Time-to-Digital Converter and Capacitance Calibration Circuit”. We have revised the manuscript according to your kind advices and referee’s detailed suggestions. Enclosed please find the responses to the referees. We sincerely hope this manuscript will be finally acceptable to be published on Micromachines. Thank you very much for your assistance and recognition of this paper.

Best regards

Sincerely yours

Jianguo Yang

Reviewer 2 Report

Language and organization are poor. 

Author Response

Dear Reviewer,

Thank you very much for your attention and the referee’s evaluation and comments on our paper “Methodology for Testing Key Parameters of Array-Level Small-Area Hafnium-Based Ferroelectric Capacitors Using Time-to-Digital Converter and Capacitance Calibration Circuit”. We have revised the manuscript according to your kind advices and referee’s detailed suggestions. Enclosed please find the responses to the referees. We sincerely hope this manuscript will be finally acceptable to be published on Micromachines. Thank you very much for all your help and looking forward to hearing from you soon.

Best regards

Sincerely yours

Jianguo Yang
